# Thin Film Composite Forward Osmosis Membrane with Single-Walled Carbon Nanotubes Interlayer for Alleviating Internal Concentration Polarization

**DOI:** 10.3390/polym12020260

**Published:** 2020-01-23

**Authors:** Yuanyuan Tang, Shan Li, Jia Xu, Congjie Gao

**Affiliations:** Key Laboratory of Marine Chemistry Theory and Technology, Ministry of Education, College of Chemistry and Chemical Engineering, Ocean University of China, Qingdao 266100, Shandong, China; 13206381095@yeah.net (Y.T.); ls251025@163.com (S.L.); gaocjie@ouc.edu.cn (C.G.)

**Keywords:** forward osmosis membrane, interlayer, single-walled nanotubes, interfacial polymerization

## Abstract

This study reported a series of thin film composite (TFC) membranes with single-walled nanotubes (SWCNTs) interlayers for the forward osmosis (FO) application. Pure SWCNTs with ultrahigh length-to-diameter ratio and without any functional group were applied to form an interconnect network interlayer via strong π-π interactions. Compared to the TFC membrane without SWCNTs interlayer, our TFC membrane with optimal SWCNTs interlayer exhibited more than three times the water permeability (*A*) of 3.3 L m^−2^h^−1^bar^−1^ in RO mode with 500 mg L^−1^ NaCl as feed solution and nearly three-fold higher FO water flux of 62.8 L m^−2^ h^−1^ in FO mode with the deionized water as feed solution and 1 M NaCl as draw solution. Meanwhile, the TFC membrane with SWCNTs interlayer exhibited significantly reduced membrane structure parameters (*S*) to immensely mitigate the effect of internal concentration polarization (ICP) in support layer with micro-sized pores in favor of higher water flux. It showed that the pure SWCNTs interlayer could be an effective strategy to apply in FO membranes.

## 1. Introduction

As a membrane separation technology, forward osmosis (FO) was operated by the natural osmotic pressure difference between low-concentrated feed solution and high-concentrated draw solution to drive water but prevent most solutes across a semi-permeable membrane [1,2]. Since the last decade, the applications of FO included, but are not limited to, wastewater purifying [3,4], brackish water and seawater desalination [5,6,7,8] and pharmaceutical applications [9,10]. Therefore, FO has turned out to be an active research area due to its potential advantages, such as low energy consumption, high water recovery and low fouling tendency [11,12]. However, the crucial challenge related to its further implementation is lack of high-performance FO membrane that both possesses high water permeability and superior water-solute selectivity [13,14].

So far, the dominating FO membrane is the thin film composite (TFC) membrane which forms a polyamide (PA) active layer on the top of a porous support layer. The PA active layer, formed by interfacial polymerization (IP), is significantly governed by the structures and properties of the support layer to determine FO membrane selectivity [15,16]. However, the typical support layers are usually the traditional polymeric ultrafiltration membranes, leading to a severe internal concentration polarization (ICP) which would cause a dramatic loss in the osmotic driving force and weaken the permeation performance. [17,18] Therefore, further enhancing water permeability with minimizing ICP effect is required to achieve ideal TFC membrane by improving the structures and properties of the PA layer and support layer.

To reduce the ICP, the support layers with large pore size such as the microfiltration (MF) membrane were used to fabricate TFC membranes, demonstrating that MF substrates could favor a high water flux and a lower *S* value [19]. However, it is still difficult to form a dense, uniform, and defect-free PA layer on the MF substrates due to a rapid and violent MPD eruption to form large initial PA oligomers and PA layer inside the substrate large pore [20]. Inspired by nanomaterials and nanotechnologies developed, carbon nanotubes (CNTs) with a hollow nano-tubular structure, an excellent chemical stability and polymer-like flexibility [21] were incorporated in the PA layer or support layer to promote the water permeability and selectivity [22,23]. However, this approach was hindered by CNTs agglomeration and CNTs/polymer incompatibility. As reported [24,25,26,27], a CNTs interlayer between the support layer with large pores and the PA layer can fine-tune the surface properties of support layer and control the IP reaction. Zhang et al. deposited graphene oxide and multiwall CNTs (GO/MWCNT) composite as an interlayer on polyethersulfone (PES) MF substrate for fabricating TFC membrane with water flux of 26.7 L m^−2^h^−1^ using 1 M NaCl as a draw solution and DI water as a feed solution [20]. Similarly, Zhou et al. fabricated TFC FO membrane with an ultrathin interlayer of polydopamine modified single-walled CNTs (PDA/SWCNTs), exhibiting the water flux of ~31 L m^−2^h^−1^ and the *S* value of 197 µm [28]. Although these TFC membrane with modified CNTs interlayer exhibited an improved FO performance, all the CNTs used were executed a hydrophilic modification, including introducing hydrated functional groups, PDA coating or blending with hydrophilic GO. It can easily damage π-π interaction to attenuate its intrinsic property. In addition, the current water flux of FO membrane due to the ICP effect was still not high. Further enhancing the water flux and relieving ICP effect should be required to achieve a target FO membrane.

In this work, in order to achieve as ideal a FO membrane as possible with an increased water flux and a lower ICP effect, a series of TFC membrane was designed with non-modified SWCNTs interlayer deposited by vacuum filtration on the mixed cellulose ester (MCE) MF substrate. The SWCNTs with an ultrahigh length-to-diameter ratio (2500–15000) was chosen due to the prefect π-π structure and enhanced mechanical properties in contrast to the functional CNTs, benefiting from forming a strong interconnected network interlayer. We believe that the SWCNTs interlayer could benefit from the formation of a high quality PA active layer. The influences of SWCNTs interlayers with various loading on the structure and transport properties of PA layer were investigated. Water flux and reverse salt flux of the fabricated membranes were also evaluated to determine the FO membrane properties and performance.

## 2. Materials and Methods

### 2.1. Materials

SWCNTs powder (diameter of 2 nm, length of 5–30 µm, purity of >95%) was purchased from XFNANO Materials Tech Co. Ltd. (Nanjing, China). The MCE MF membrane with a mean pore size of 0.22 µm was obtained from Beijing Shenghe Membrane Tech Co., Ltd. (Beijing, China). Sodium dodecylbenzenesulfonate (SDBS, technical grade), m-phenylenediamine (MPD) (>99.0%) and trimesoyl chloride (TMC) (>98.0%) were purchased from Sigma-Aldrich Chemical Co. Ltd. (Burlington, MA, USA). Other chemicals were obtained from Sinopharm Chemical Reagent Co. Ltd. (Shanghai, China) and were used without further purification. The deionized water used for solution preparation and experimental washing was produced by a Millipore water purification system with the resistivity of 18.0 MΩ cm^−1^.

### 2.2. Membrane Preparation

The SWCNTs dispersion was prepared as the detailed procedure described in the previous work [29,30]. In details, SWCNTs powder and SDBS powder was mixed concurrently in water and sonicated for 10 h under a power of 300 W to obtain SWCNTs dispersion of 0.1 g L^−1^. Subsequently, the dispersion was centrifuged at 10,000 rpm for 0.5 h twice to collect supernatant as final SWCNTs dispersion with an actual SWCNTs concentration of ~0.067 g L^−1^. The SWCNTs interlayer was prepared by vacuum filtration of the quantitative SWCNTs dispersion on the MCE membrane surface with the diameter of 4.0 cm at room temperature (25 °C) and under an absolute pressure of −0.07 MPa. The corresponding substrates were donated as MCE/CNTs_X_, where subscript x was denoted the volume (mL) of SWCNTs suspension deposited in the range of 0–10 mL and the loading of SWCNTs interlayer was calculated.

The PA active layer was prepared via conventional interfacial polymerization as previous literature [31]. The SWCNTs interlayer on the top of MCE membrane was first contacted with 2 wt% MPD aqueous solution for 2 min and the residual aqueous drops were removed by a rubber roller. Afterwards, the 0.1 *w*/*v*% TMC dissolved in n-hexane solution was poured onto the SWCNTs interlayer surface reacting for 30 s to form a nanoscale polyamide active layer. After draining the excess TMC solution, the membrane was washed by n-hexane and heated treating in an oven at 80 °C for 1 min, and then the resultant TFC membranes were kept in DI water before test. In contrast, the MCE membrane was used as support layer individually to prepare TFC membrane as control group. The fabrication procedure of the TFC membrane was described in Figure 1. The abbreviation TFC_X_ was on behalf of prepared TFC FO membranes, where subscript x was denoted the volume of SWCNTs suspension deposited.

### 2.3. Characterization

The surface and cross-sectional morphologies of SWCNTs interlayer and TFC membrane were characterized by scanning electron microscopy (SEM, S-4800, Hitachi, Japan). Samples were dried overnight and gold sputtering was performed on the sample before observation. The cross-sectional morphologies were also obtained by transmission electron microscope (TEM, H-7650, Hitachi, Japan) at an acceleration voltage of 100 kV. The image was also analyzed with the software ImageJ to determine the thicknesses of the bald SWCNTs interlayer and the PA active layer. Meanwhile, the thickness of SWCNTs interlayer was measured using a surface profiler (Dektak 150, Veeco, Plainview, NY, USA). The surface hydrophilicity was measured with a water contact angle (WCA) measuring system (DSA 100, Kruss, Hamburg, Germany) in six random positions to acquire the average value. The pure water permeability was carried out using a dead-end filtration system to evaluate the permeability of SWCNTs interlayers. The surface zeta potential was evaluated using 0.005 mol L^−1^ KCl aqueous solution at pH 7.0 at room temperature by an electrokinetic analyzer (SF-SA, Saifei, Hangzhou, China) [26]. The mechanical properties including tensile strength, Young’s modulus and elongation at break were measured with a sample size of 1 cm × 0.5 cm by the tensile testing equipment (QJ210, Aike Technology, Shanghai, China) according to an ASTM method. The element composition and chemical bonding of the TFC membranes were analyzed by X-ray photoelectron spectroscopy (XPS, Thermo ESCALAB 250XI, VG Scientific, Waltham, MA, USA).

### 2.4. Intrinsic Transport Property and FO Performance Measurement

The intrinsic transport properties including pure water permeability coefficient (*A*, L m^−2^ h^−1^ bar^−1^), salt rejection (*R*, %) and salt permeability coefficient (*B*, L m^−2^ h^−1^) were evaluated under a lab-scale cross-flow RO test [32]. All the membranes were pre-compacted for 1.0 h under the pressure of 3.0 bar to acquire the stable value. Each membrane sample was tested for at least three times to obtain the average value, and the error bar represented the standard deviation. The pure water permeability coefficient was determined from pure water flux (25 °C) under trans-membrane pressure (Δ*P*) of 2.0 bar according to Equation (1). The effective area (*A*_m_) of membrane sample was 11.3 cm^2^. *B* value was calculated based on the average rejection value (3 replicates) at a given pressure by 500 mg L^−1^ NaCl solution as feed solution [33] according to Equations (2) and (3).
(1)$$A=\frac{J_{w}}{∆P}=\frac{∆V}{A_{m}∆t∆P}$$
(2)$$R=\left(1-\frac{C_{p}}{C_{f}}\right)$$
(3)$$\frac{1-R}{R}=\frac{B}{A\left(∆P-∆\pi \right)}$$
where Δ*V* (L) was the volume of permeate water collected over the operating time Δ*t* (h). *C**_f_* and *C**_p_* were the concentrations of feed solution and permeate solution, respectively. In addition, Δ*π* was the osmotic pressure difference.

The FO performance of prepared TFC membrane was evaluated using lab-scale cross-flow unit in FO mode where the active layer was oriented towards the feed solution including water flux (*J**_v_*, L m^−2^ h^−1^) and reverse salt flux (*J**_s_*, g m^−2^ h^−1^) described as previous study [34]. During the FO tests, DI water was used as feed solution and 1 M NaCl solution was used as draw solution. The average cross-flow velocity was 6.3 cm s^−1^ and the effective membrane area (*A**_m_*) was 12.56 cm^2^. Each membrane sample was tested for at least three times to obtain the average value, and the error bar represented the standard deviation. The water flux was determined by measuring the weight change of the feed solution with electronic balance (CP1502, Ohaus, Shanghai, China) according to Equation (4). The reverse salt flux was detected by the salt concentrations change in the feed solution using the conductivity meter (DDS-307A, Rex Electric Chemical, China) according to Equation (5) [14,35].
(4)$$J_{v}=\frac{∆V_{t}}{A_{m}∆t}$$
(5)$$J_{s}=\frac{C_{t}V_{t}-C_{0}V_{0}}{A_{m}∆t}$$
where Δ*V**_f_* (L) was the volume change of feed solution that has permeated across the membrane in a predetermined time Δ*t* (h) during the tests. *C*_0_ and *V*_0_ were the initial NaCl concentration (g L^−1^) and the volume (L) of the feed solution, while *C*_t_ and *V*_t_ were the NaCl concentration and the volume after predetermined time Δ*t* (h), respectively.

The structural parameter (*S*, µm) of the FO membrane can be determined to characterize the effect of ICP, which can be calculated by the following equation in FO mode [36].
(6)$$s=\frac{D}{J_{v}}\left[\ln\frac{A\pi_{draw}+B}{A\pi_{feed}+J_{v}+B}\right]$$
where *D*, *π**_draw_* and *π**_feed_* were the solute diffusion coefficient in water and the osmotic pressure of the draw solution and feed solution, respectively.

## 3. Results

### 3.1. SWCNTs Interlayer Characterization

An optimum membrane preparation process is essential for achieving high-efficient TFC membrane. The schematic diagram of TFC membrane preparation process in this work is shown in Figure 1. For comparison, a typical TFC membrane without SWCNTs interlayer exhibited two-tier structure with PA layer formed on the MCE substrate directly (Figure 1a). With the presence of SWCNTs interlayer, the TFC membrane possessed a typical three-tier structure with a top PA layer, a middle SWCNTs interlayer and a bottom MCE substrate (Figure 1b).

Properties of MCE/CNTs support layer including pure water permeance, thickness, hydrophilicity and morphology were investigated and the results are shown in Figure 2 and Figure 3. The SWCNTs loading could be calculated from the volume of the SWCNTs dispersion solution due to the prefect retention of all the SWCNTs with ultra-large length-to-diameter ratio on the MCE substrate. The permeation properties of the support layer are critical in determining the water permeation of TFC membrane [37]. From Figure 2a, the presence of SWCNTs interlayer triggered a decrease in the water permeance of MCE/CNTs support layer compared to the pristine MCE substrate, mainly ascribing to following reasons: (1) the increasing thickness of SWCNTs interlayer (Figure 2b); (2) the severe hydrophobicity with a contact angle of 110–115° (Figure 2b), and (3) the diminished surface pore size (Figure 3a,c) leading to the elevated water transport resistance. With the increasing SWCNTs loading from 0.27 g m^−2^ to 0.53 g m^−2^, the thickness of MCE/CNTs support layer increased from 38 to 373 nm, while the contact angle almost kept constant. The descending water permeance was mainly dominated by the thickness and surface pore size. It was noteworthy that all the MCE/CNTs support layers exhibited a higher water permeance than those commercial and on-going reported FO support layers [15,16,38,39,40], demonstrating an enormous potential for developing high-flux FO membrane.

As reported, the surface morphology of support layer plays an important role in PA layer formation for TFC membrane [41]. The top surface SEM images of MCE/CNTs layer were showed in Figure 3a,c. For MCE/CNTs_1_ support layer, the disordered SWCNTs interconnected networks partly covered the micron-size pores of MCE substrate. With an increasing SWCNTs loading, a continuous SWCNTs networks completely covered the MCE substrate and a denser surface was obtained, also confirmed by the AFM images (Figure 3b and Appendix A). From the cross-sectional images of the MCE/CNTs layer (Appendix A), the SWCNTs interlayer was found to be tiled on the top of MCE substrate. As the SWCNTs loading increased, the distinct double-layer structure consisted of a uniform and even SWCNTs interlayer and MCE substrate was observed. To further recognize the cross-sectional structure of SWCNTs interlayer, it was separated from MCE substrate and put on the silicon wafer prior to the SEM measurement. Appendix A exhibited the entire and bulk inter-network structure of typical SWCNTs, indicating the total retention of SWCNTs dispersion solution on the MCE substrate.

### 3.2. TFC FO Membrane Characterization

Our TFC membrane was comprised of PA active layer interfacially polymerized on the top of SWCNTs interlayer. The effect of SWCNTs interlayer on the morphologies of PA layer was presented in Figure 4. From Figure 4a, granular structures with unevenly large leaf-like structures were observed on the TFC_0_ membrane surface, which was different from the typical leaf-like structures for the TFC membranes with SWCNTs interlayer. This observation was related to larger pores on the MCE substrate, in which MPD can rapidly diffuse and react with TMC to form large PA oligomers and an initial PA layer [20,42]. The denser surface of MCE/CNTs support layer (Figure 3a) favored the more uniform leaf-like structure of the PA layer, attributing that the SWCNTs interlayer can make the MPD diffusion slow and uniform from aqueous solution to organic solution and thus control the interfacial polymerization reaction [43]. Cross-sectional SEM images (Appendix A) indicated a continuous PA layer formed on the support layer surface for all the TFC membranes. Compared to the membrane with MCE substrate, a flat PA layer was found for the membrane with the MCE/CNTs support layer due to the relatively flat SWCNTs interlayer. The similar difference in the cross-sectional morphology was further found by TEM (Figure 4b). In addition, the thickness of PA layer reduced significantly from ~154 nm (TFC_0_ membrane) to ~70 nm (TFC_1_ membrane), which can shorten the water passage and thus drop the water transport resistance. So it was proved that the SWCNTs interlayer had a remarkable influence on the structure of PA active layer, which might be favorable for an improvement in the separation performance.

Due to a crucial role of surface charge in the separation performance according to Donnan exclusion theory and dielectric effects for salt rejection and selectivity [44], surface zeta potentials of all the TFC membranes and support layers were investigated and presented in Figure 5a. The pristine MCE substrate surface was negatively charged due to its intrinsic property. When the SWCNTs interlayer covered the MCE substrate, there was almost no surface charge due to the pure SWCNTs without functional groups. For all the TFC membranes, the surfaces were negatively charged because of the deprotonation of carboxyl groups arising from the unreacted acyl chloride hydrolysis. The negative charge increased from −18.1 mV of TFC_1_ membrane to −30.5 mV of TFC_10_ membrane, probably indicating more insufficient IP reaction with the SWCNTs loading. Besides, effect of SWCNTs interlayer on the mechanical properties of membranes was illustrated in Figure 5b. No matter support layers or TFC membranes, the mechanical stress and Young’s modulus was reinforced gradually as the SWCNTs loading increased. Moreover, compared to the MCE/CNTs support layer, the corresponding elongation at break for TFC membranes was intensified thanks to the flexible PA layer.

XPS analysis was carried out to evaluate the elemental composition and chemical bonding in the top 5–10 nm of the PA active layer [45]. Three distinct peaks of C 1s, N 1s, and O 1s were observed in the XPS survey spectra for all the prepared TFC membranes (Appendix A). To quantify chemical species within PA layer, the high-resolution XPS spectra was exhibited in Figure 6. The core level O 1s spectrum was primarily split into two peaks: N-C=O at 531.5 eV ascribing to the amide bond deriving from the reaction between the amino group and acyl chloride group and O-C=O at 532.8 eV ascribing to carboxylic acid group deriving from the hydrolysis of unreacted acyl chloride group in TMC molecule [25]. The increasing proportion of O−C=O/N-C=O indicated the more unreacted acyl chloride and less complete reaction between MPD and TMC, which can character a lower cross-linking degree of the PA layer [33]. Therefore, the prepared PA layer was less dense with the increasing SWCNTs loading (Appendix A) in accordance with the results measured by zeta potential (Figure 5a), which was attributed that serious hydrophobicity and increased thickness were detrimental for storage and migration of MPD. Besides, the core level N 1s spectrum was split into two peaks including the amide bond at 400.0 eV (N-C=O) and a small contribution of unreacted amino group at 398.5 eV (-NH_2_) or 401.7 eV (-N*H_3_) [27,33]. Moreover, -NH_2_ can be completely converted into -N*H_3_ when introducing higher SWCNTs loading, relevant to the protonation of the more residual -NH_2_ in MPD molecule resulting from more carboxylic acid group. The results revealed that high SWCNTs loading may lower the cross-linking degree of the PA layer, which was adverse to the improvement of membrane selectivity.

### 3.3. Intrinsic Separation Properties and FO Performance of TFC Membranes

As mentioned before, SWCNTs interlayer had a significant effect on the structure and properties of PA active layer. Meanwhile it also played an important role in enhancing membrane performance. The separation performance of the prepared TFC membrane was investigated and the results were presented in Figure 7. According to intrinsic separation properties measured in RO mode (Figure 7a), the SWCNTs interlayer motivated enhancement of water permeability (*A*) of TFC membranes, ascribed that the thinner, uniform and even PA active layer (Figure 4) supplied less water transport resistance. Especially for TFC_5_ membrane, its water permeability coefficient (3.3 L m^−2^h^−1^bar^−1^) was more than 3.5-fold higher than that of control TFC_0_ membrane (0.9 L m^−2^h^−1^bar^−1^). However, for TFC_10_ membrane, the *A* value began to decline, consistent to the water permeance of MCE/CNTs support layer (Figure 2a). The similar trend of water flux (*J_v_*) measured in FO mode can be proved in Figure 7b. In details, for the optimized TFC_5_ membrane, its water flux increased from the pristine 23.5 to 62.8 L m^−2^ h^−1^, improved by 167%. It could be concluded that it was the structure and properties of PA layer and support layer that collectively decided the water transport for the TFC membranes. In addition, the salt permeability (*B*) and reverse salt flux (*J_s_*) of TFC membrane increased after the introduction of SWCNTs interlayer (Figure 7a,b) resulting from lower cross-linking degree of PA active layer demonstrated by XPS results (Figure 6). However, as the SWCNTs loading increased, the reverse salt flux almost maintained stable. It might be inferred that it was the contact hydrophobicity rather than increasing thickness of SWCNTs interlayer (Figure 2b) that had a main influence on the reverse salt flux. Usually, the reverse salt diffusion exerted a decreasing osmotic driving force, causing the loss of the water flux discussed in FO process. However, for prepared TFC membrane, the water flux was incremental while the reverse salt flux maintained stable, which might break the trade-off phenomenon occurred in most of the polymer membranes. The specific salt flux (*J**_s_/J_v_*), the lost amount of draw solute per water unit, is a practical indicator for evaluating the selectivity of FO membranes [46]. Unexpectedly, the *J**_s_/J_v_* showed no conspicuous improvements after introduction of SWCNTs interlayer. Although the lowest *J**_s_/J_v_* value of 0.3 g L^−1^ was presented for TFC_5_ membrane (Figure 7b), showing a relatively enhanced separation selectivity among all the prepared membranes, this value was unsatisfactory compared to those reported in the literature [28,47,48]. In conclusion, the SWCNTs interlayer benefited to improve water flux, yet was not favorable for the superior selectivity which needed to be further improved by hydrophilization in the premise of no harming its inherent structure [28,49].

Membrane structure parameter (*S*) is a direct characterization of ICP effect, which has units of length and can be thought as the distance that a solute particle must travel from the bulk draw solution to the membrane active layer [36]. In Figure 7c, all the TFC membranes with the SWCNTs interlayer presented the lower *S* value compared to the TFC_0_ membrane, attributing that the interconnected SWCNTs with ultrahigh length-to-diameter ratio endowed the low tortuosity. The lowest *S* value of only 88 µm was appeared on TFC_5_ membrane, which represented the weakest ICP effect supporting the highest water flux. Notably, the *S* value of prepared TFC membrane was lower than those reported in most literature [21,50,51,52], demonstrating that the SWCNTs interlayer was of great significance to prepare high-flux FO membrane. The variation trend of the time-dependent normalized water flux can also show the effect of ICP on separation performance of FO membrane [49]. As presented in Figure 7d, after 4 h test, the TFC membranes with the SWCNTs interlayer showed smaller water flux decline with less than 40% of its original flux, while the TFC_0_ membrane seriously decreased by 58%. Obviously, the existence of the SWCNTs interlayer can alleviate ICP effect in macro-pore support layer. Besides, the least decline of water flux was obtained for TFC_5_ membrane, conforming to the highest water flux and lowest *S* value.

### 3.4. Comparisons of Intrinsic Separation Properties and FO Performance

The performance of prepared TFC membranes with SWCNTs interlayers in this work was compared with that of reported TFC membrane with other interlayers. The result was presented in Table 1. For a visual comparison, the selected membranes were all tested in FO mode with 1M NaCl as draw solution and DI water as feed solution except for the special annotation. The prepared TFC membrane with a SWCNTs interlayer in this work exhibited the more than two folds higher water flux and less than half the lower *S* value than in other studies, indicating the least effect of ICP. In addition, the prepared TFC membranes also exhibited the competitive water permeability and *A*/*B* value. However, meanwhile, the higher specific salt flux occurred on the prepared TFC membrane, implying the unsatisfactory selectivity, which needed to be improved in future work in order to acquire high-performance FO membrane with both high water permeability and superior water-solute selectivity.

## 4. Conclusions

In this study, a series of TFC FO membranes with SWCNTs interlayers and without any functional group was developed. The effects of a SWCNTs interlayer on the TFC membrane properties and performance were investigated. The TFC membranes with the optimal SWCNTs interlayer showed significantly enhanced water flux and superior *S* value, indicating that SWCNTs interlayers were beneficial to alleviate effect of ICP. However, the selectivity of the prepared TFC membranes was less competitive than that reported in the literature due to the less cross-linking degree on PA layer related to the hydrophobic SWCNTs interlayer surface. Therefore, it needs urgent solutions to enhance selectivity of resultant TFC membrane for example the hydrophilization of SWCNTs interlayer. Finally, study suggested that the SWCNTs interlayer would be an effective strategy to apply in FO process.

## Figures and Tables

**Figure 1 polymers-12-00260-f001:**
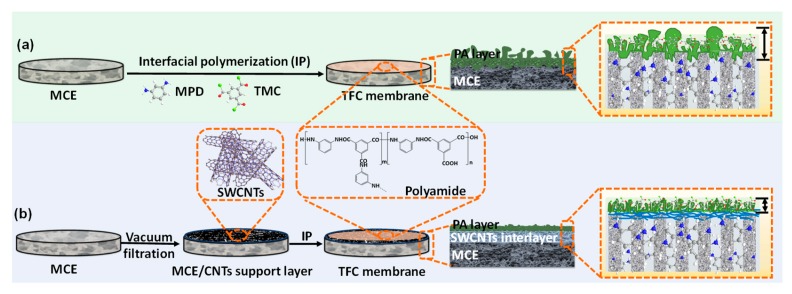
Schematic diagram of preparation process for TFC membrane. (**a**) TFC membranes with pristine MCE substrates; (**b**) TFC membranes with MCE/CNTs support layers.

**Figure 2 polymers-12-00260-f002:**
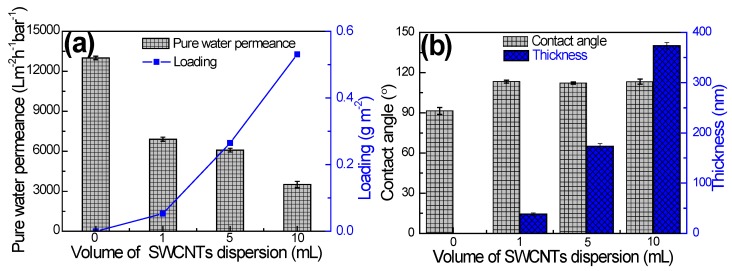
(**a**) Pure water permeance and (**b**) contact angle and thickness of MCE/CNTs support layer with volume of SWCNTs dispersion solution.

**Figure 3 polymers-12-00260-f003:**
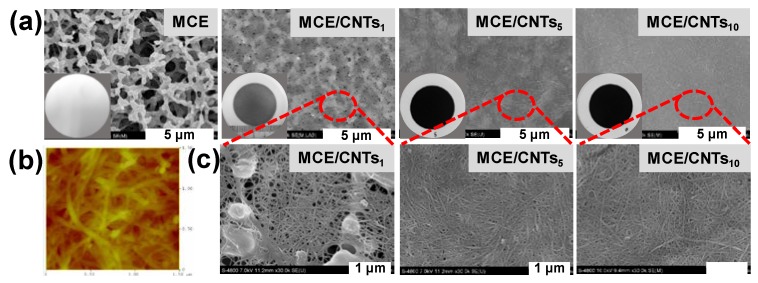
Surface morphology of MCE/CNTs support layer. (**a**) surface SEM images with insets of digital photos, (**b**) surface AFM morphology of MCE/CNTs_10_ support layer and (**c**) magnified surface SEM images.

**Figure 4 polymers-12-00260-f004:**
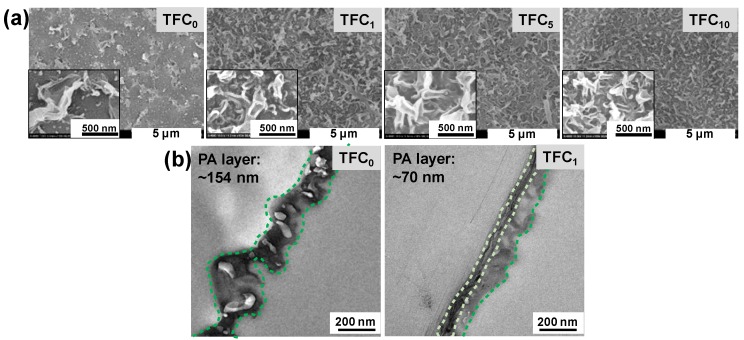
Morphology characterization **of** TFC membranes with SWCNTs interlayer. (**a**) surface SEM images with insets of images magnification × 10.0 K and (**b**) cross-sectional TEM images of TFC_0_ and TFC_1_ membranes.

**Figure 5 polymers-12-00260-f005:**
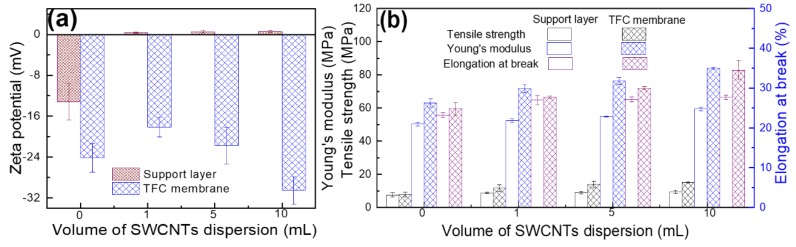
(**a**) Surface zeta potential and (**b**) mechanical properties including tensile strength, Young’s modulus and elongation at break.

**Figure 6 polymers-12-00260-f006:**
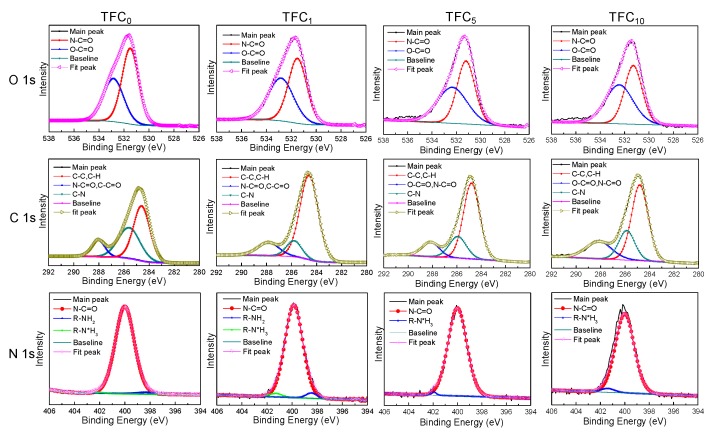
High-resolution XPS spectra of TFC membranes.

**Figure 7 polymers-12-00260-f007:**
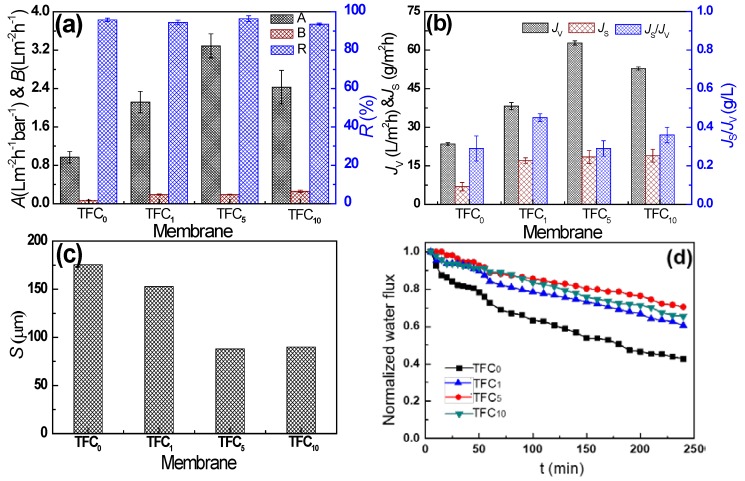
Separation performance of prepared TFC membrane with various SWCNTs interlayer. (**a**) Intrinsic separation properties including water permeability coefficient (*A*), salt permeability coefficient (*B*) and salt rejection (*R*), (**b**) FO performance including water flux (*J**_v_*), reverse salt flux (*J**_s_*) and the ratio of reverse salt flux to water flux (*J**_s_/J_v_*), (**c**) membrane structural parameter (*S*) and (**d**) time-dependent normalized water flux.

**Table 1 polymers-12-00260-t001:** Summarizes the FO performance and intrinsic separation properties of FO membranes with interlayers reported in the recent literature.

Membrane	*J_v_* (L m^−2^ h^−1^)	*J_s_* (g m^−2^h^−1^)	*J_s_/J_v_* (g L^−1^)	*A* (L m^−2^h^-1^bar^−1^)	*B* (L m^−2^ h^−1^)	*A*/*B* (bar^−1^)	*S* (µm)	Ref.
TFC membrane with SWCNTs interlayer on MCE MF membrane	62.8	19.4	0.29	3.3 ^c^	0.19 ^c^	17.3	88 ^c^	This work
TFC membrane with PDA coated CNTs interlayer on PES MF support layer	31.0	0.6	0.02	2.0 ^d^	0.05 ^d^	39.0	197 ^d^	[28]
RGO layer on CNTs hollow fiber support layer	22.6 ^a^	1.6 ^a^	0.07	2.1 ^e^	0.05 ^e^	41.4	202 ^e^	[47]
TFC membrane with interlayer decorated metal−organic framework UiO-66 on PSf support layer	11.0	2.9	0.27	4.5 ^f^	0.81 ^f^	5.5	741 ^f^	[48]
TFC membrane with CNTs interlayer with carboxyl groups on PVDF support layer	24.0 ^b^	5.9 ^b^	0.25	1.3 ^g^	0.54 ^g^	2.3	392 ^g^	[53]
TFC membrane with GO/MWCNTs interlayer on PES MF support layer	17.2	3.7	0.22					[27]
TFC membrane with PDA/GO interlayer on PSf support layer	24.3	3.8	0.16					[49]
TFC membrane with PDA/ halloysite nanotubes (HNT) interlayer on PSf support layer	26.9	4.0	0.15					[54]
TFC membrane with GO interlayer on PVDF support layer	17.5	1.0	0.06					[55]

^a^ Data estimated with 0.5 M NaCl as draw solution. ^b^ Data estimated with 2.0 M NaCl as draw solution. ^c^ Data estimated with 500 mg L^−1^ NaCl solution as feed solution at a pressure of 2.0 bar. ^d^ Calculated by the Excel-based error minimization algorithm developed by Tiraferri et al. [56]. ^e^ Data estimated with 500 mg L^−1^ NaCl solution as feed solution at a pressure of 1.0 bar. ^f^ Data estimated with 1000 mg L^−1^ NaCl solution as feed solution at a pressure of 2.0 bar. ^g^ Data estimated with 10 mM NaCl solution as feed solution at a pressure of 1.0 bar.

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
