# Peer review of "Thin Film Composite Forward Osmosis Membrane with Single-Walled Carbon Nanotubes Interlayer for Alleviating Internal Concentration Polarization"

_polymers, 2020, doi:10.3390/polym12020260_

Round 1
Reviewer 1 Report
In this work, a series of thin film composite (TFC) membranes with single‐walled nanotubes (SWCNTs) interlayers for the forward osmosis (FO) application was prepared, characterized and investigated. It was determined that the TFC membranes with optimal SWCNTs interlayer significantly enhanced water flux and showed a superior membrane structure parameter (S) value, however, the selectivity of the prepared TFC membranes was not so good as expected.
The work done by the authors is very interesting and well described. My comments and questions for the authors:
The authors stated that the cross‐linking degree of PA layers differ according to the data of XPS spectra. I would suggest to determine the cross-linking degree of PA layers by additional methods for full characterization and confirmation. How do the authors suggest to increase the hydrophilicity of SWCNTs interlayer surface which caused the poor selectivity of TFC membranes?Author Response
Please see the attachment.

Reviewer 2 Report
The manuscript is quite interesting, however there are some points that need to be improved or cleared.
For instance, in line 100, the authors indicate that reaction of polyamide is carried out, is it the long enough time to form the polyamide 30 seconds? And how can be proved this?
In line 120, it is indicated that mechanical properties of materials were measured, however there a lot of mechanical properties, it is needed to indicate which kind of properties were determinate. Also it is needed to indicate if this kind of test were evaluated according with an ASTM method.
According with figure 1, was the interfacial polymerization carried out for materials illustrated in fig 1a and 1b?
It is not clear how the increasing of thickness can has influence over the contact angle and hydrophobilicyt of MCE/CNTs, if contact angle has not significative change from 1 to 10 volume of SWCNTs dispersion but thickness changes from around 50 to almost 400 nm.
Th figure 2c has not much sense to be presented, it would be better to include AFM micrographs than these SEM images.
Lines 242-244, figure 3c, indicates that the core Level N1s spectrum split into two peaks, associated to amide bond and amino group, however in figure that it is not displayed, only one peak it is observed.
Lines 246-247, authors indicate that crosslinking degree of PA layer decrease, how can demonstrate that crosslinking of PA has a decreasing.
Authors want to show a lot of information, however this can be confuse to reader and create confusion in that way, so it is needed to re-distribute the information presented in figures 2 to 5.
The information displayed in table 1 I consider that can be discussed in text instead of the table.
Round 2
Reviewer 2 Report
After review in the manuscript, the authors correct or clarify the comments done by reviewer, the only question is about the information presented in figures 2 to 5, the recommendation about separate the information displayed in those figures is as follows:
may be figure 2, can be separated as: figures (micrographs) in one and contact angle and pure water permeance. for figure 3, separate volume of SWCNTs from micrographs for figures 4 and 5, those are ok.This is the only recommendation before to consider the manuscript can be accepted.
Author Response
Response to Reviewer :
After review in the manuscript, the authors correct or clarify the comments done by reviewer, the only question is about the information presented in figures 2 to 5, the recommendation about separate the information displayed in those figures is as follows:
may be figure 2, can be separated as: figures (micrographs) in one and contact angle and pure water permeance. for figure 3, separate volume of SWCNTs from micrographs for figures 4 and 5, those are ok.
Response:
Thank you very much for these recommendations. Authors totally agree with the reviewer. We have re-distributed these figures, and the numbers of these Figures have been corrected in the manuscript.
For pristine Figure 2, we have divided it into the two Figures:
Figure 2. (a) Pure water permeance and (b) contact angle and thickness of MCE/CNTs support layer with volume of SWCNTs dispersion solution.
Figure 3. Surface morphology of MCE/CNTs support layer. (a) surface SEM images with insets of digital photos, (b) surface AFM morphology of MCE/CNTs10 support layer and (c) magnified surface SEM images.
For pristine Figure 3, we also have divided it into the two Figures:
Figure 4. Morphology characterization of TFC membranes with SWCNTs interlayer. (a) surface SEM images with insets of images magnification ×10.0 K and (b) cross-sectional TEM images of TFC0 and TFC1 membranes.
Figure 5. (a) Surface zeta potential and (b) mechanical properties including tensile strength, Young’s modulus and elongation at break.
